# Detection of QTLs for Plant Height Architecture Traits in Rice (*Oryza sativa* L.) by Association Mapping and the RSTEP-LRT Method

**DOI:** 10.3390/plants11070999

**Published:** 2022-04-06

**Authors:** Hélder Manuel Sitoe, Yuanqing Zhang, Siqi Chen, Yulong Li, Mehtab Ali, Ognigamal Sowadan, Benjamin Karikari, Erbao Liu, Xiaojing Dang, Hujun Qian, Delin Hong

**Affiliations:** 1State Key Laboratory of Crop Genetics and Germplasm Enhancement, Nanjing Agricultural University, Nanjing 210095, China; heldermsitoe@gmail.com (H.M.S.); m15110403618@163.com (Y.Z.); 2018101067@njau.edu.cn (S.C.); 2018101068@njau.edu.cn (Y.L.); mehtabalirajpar@gmail.com (M.A.); ognigamalsowadan@yahoo.fr (O.S.); benkarikari1@gmail.com (B.K.); liuerbao@ahau.edu.cn (E.L.); dangxj@aaas.org.cn (X.D.); qianhujun@njau.edu.cn (H.Q.); 2Faculty of Agronomy and Biological Sciences, Púnguè University, P.O. Box 323, Manica 2202, Mozambique; 3Department of Crop Science, Faculty of Agriculture, Food and Consumer Sciences, University for Development Studies, P.O. Box TL 1882, Tamale 00233, Ghana; 4College of Agriculture, Anhui Agricultural University, Hefei 230031, China; 5Institute of Rice Research, Anhui Academy of Agricultural Sciences, Hefei 230031, China

**Keywords:** plant height and component traits, favorable alleles, elongated internodes, chromosome segment substitution lines, natural population, *Oryza sativa*

## Abstract

Plant height (PH) and its component traits are critical determinants of lodging resistance and strongly influence yield in rice. The genetic architecture of PH and its component traits were mined in two mapping populations. In the natural population composed of 504 accessions, a total of forty simple sequence repeat (SSR) markers associated with PH and its component traits were detected across two environments via association mapping. Allele RM305-210 bp on chromosome 5 for PH had the largest phenotypic effect value (PEV) (−51.42 cm) with a reducing effect. Allele RM3533-220 bp on chromosome 9 for panicle length and allele RM264-120 bp on chromosome 8 for the length of upper first elongated internode (1IN) showed the highest positive PEV. Among the elongated internodes with negative effects being desirable, the allele RM348-130 bp showed the largest PEV (−7.48 cm) for the length of upper second elongated internode. In the chromosome segment substitution line population consisting of 53 lines, a total of nine QTLs were detected across two environments, with the phenotypic variance explained (PVE) ranging 10.07–28.42%. Among the detected QTLs, *q1IN-7* explained the largest PVE (28.42%) for the 1IN, with an additive of 5.31 cm. The favorable allele RM257-125 bp on chromosome 9 for the 1IN increasing was detected in both populations. The favorable alleles provided here could be used to shape PH architecture against lodging.

## 1. Introduction

Rice (*Oryza sativa* L.) is one of the most important staple crops serving as a major food source for over 3.5 billion people worldwide; hence, it is considered to be a food security crop worldwide [1,2]. The world rice cultivation acreage was 167.13 million hectares and the production of milled rice was 509 metric tons in year 2018/2019. China was the leading producer, with 148.5 million metric tons of milled rice on the paddy field of 31 million hectares, followed by India, with 116.42 million metric tons on the rice acreage of 40 million hectares [3]. The global population is growing continuously, and the arable land acreage is continuously reducing, creating threats to food security. It has been estimated that rice production needs to increase by more than 42% of the current production in order to meet future needs by 2050 [4]. It is imperative to increase the yield of rice per unit area through genetic improvement to meet the food demands of this growing global population.

The use of semi-dwarf rice cultivars solves the problem of increasing fertilizer application without lodging before harvest, so as to increase yield (compared with high stalk cultivars) [5,6]. The use of hybrid rice cultivars can increase yield (compared with semi dwarf conventional cultivars) by using heterosis and appropriately increasing total biomass [7,8]. However, in order to further improve the grain yield per unit area of hybrid rice varieties, it is necessary to manipulate the length of each internode in the aboveground part of rice plants, so as to make the field plant population develop to the upper space, and not only increase the biomass per unit area, but also not lead to lodging before harvest. This plant height design is also necessary for development of the parents of hybrid rice cultivars (male sterile line, maintainer line, restorer line and photo-thermosensitive genic male sterile line) and conventional cultivars.

In general, the plant height (PH) in rice plant comprises four elongated internodes on the main stem. So far, at least thirty-six quantitative trait loci (QTLs) have been identified and reported for PH in rice, including 3, 4, 4, 1, 2, 4, 3, 6, 2, 4, 2 and 1 QTL on chromosomes 1 to 12, respectively [9,10,11,12]. The phenotypic variance explained (PVE) by these QTLs ranged from 2.41 to 28.05%, and of these, 8 of the 36 QTLs have a PVE >10%. Some of the genes for PH have been cloned, and the functions have been elucidated; for example, *CYP94C2b* is a gene controlling PH and the elongated internode lengths, and the overexpression of *CYP94C2b* increased the height and the internode lengths [13]. Another candidate gene, *OsMPH1*, has also been demonstrated to control PH; the overexpression of *OsMPH1* leads to increases in PH and grain yield in rice [14].

Panicle length (PL) is a quantitative trait controlled by both major and minor QTLs and influence by environment [15]. To date, at least 253 QTLs for PL have been reported across the 12 chromosomes [15,16,17,18,19]. However, only a few genes regulating panicle development have been cloned and functionally validated, such as *LP1* [19], *DEP2* [20] and *DEP3* [21] over the past two decades. Additionally, TUT1 is another gene encoding a functional SCAR/WAVE protein which plays an essential role in regulating F-actin organization, functioning in multiple aspects of rice growing stages, especially in panicle development [22].

For the top first internode, four QTLs were detected on chromosomes 3, 7, 8 and 9, explaining 20.3% of the total phenotypic variation [23]. Until today, several studies have cloned and elucidated the function of the elongated uppermost internode gene, designated as EUI1, which is characterized by the near doubling 1IN [24,25,26].

For the 2IN, seven QTLs were mapped on seven chromosomes and collectively accounted for 34.1% phenotypic variation [23]. Wu, et al. [27] reported that a dm-type rice mutant *DMF-1* previously induced by X-ray irradiation of plants on the rice cultivar Fujiminori revealed semi-dwarfism and high lodging resistance with an apparently normal growth, but with inhibition of elongation of the 2IN, being controlled by a dominant gene *Ssi1* (Short second internode 1). A QTL flanked by OSR7 and RM230 on chromosome 8 had important effects on the first and second elongated internodes.

For the 3IN, six QTLs mapped on chromosomes 1, 6, 8, 10 and 12 caused 29.0% phenotypic variation for this trait. Six QTLs for the top fourth internode (4IN) were detected on chromosomes 1, 3, 4, 6, 10, and 12, which collectively explained 39.6% phenotypic variation [23]. The semi-dwarfing gene, *sd-1*, contributes significantly to lodging resistance, inhibiting the elongation of lower internodes, particularly the 3IN and 4IN, more than upper internodes, because lodging often takes place at the lower internodes [28]. A study on short basal internodes reported that a rice semi-dominant lodging resistance gene, *SBI* (shortened basal internodes), encodes a culm-specific expressed *OsGA2ox* that reduces the culm content of bioactive gibberellin, particularly in the basal internodes [29]. In addition, higher activity of the *SBI* significantly reduced the rice PH, and, therefore, its application in rice production contributed to breed improved lodging-resistant rice varieties and thus increased yield. Although several studies on associations between markers and the plant height have been conducted in rice [10,30,31,32,33], to date, no report has documented a QTL analysis of the relationships between plant height and its components traits (panicle length and the length of all internodes), using two mapping strategies.

The objectives of the present study were (i) to map QTLs for PH and its component traits in a natural population consisting of 504 rice accessions, using a genome-wide association study (GWAS) mapping strategy; (ii) to detect QTLs for PH and its component traits in a chromosome segment substitution line population designated XC-CSSLs population by likelihood ratio test based on stepwise regression (RSTEP-LRT) method; (iii) to identify favorable marker alleles associated with PH and its component traits in the two populations; and (iv) to predict parental combinations that could be used to improve PH and its component traits by pyramiding favorable alleles method for breeding cultivars with a high yield and resistance to lodging.

## 2. Results

### 2.1. Phenotypic Evaluations of the Six Traits in the Natural Population Composed of 504 Rice Accessions

Table 1 shows descriptive statistics for each trait calculated from the 504 accessions in 2018 and 2019. The phenotypic data of most of the traits largely followed a normal continuous distribution that is typical of quantitative traits (Figure 1).

The analysis of variance (ANOVA) showed highly significant (*p* < 0.01) differences among the 504 accessions for each trait, indicating that there exists high genetic variation in the natural population. There was no statistically significant difference between 2018 and 2019 in means of the entire population in each trait. In 2018, the PH among the 504 accessions ranged from 68.78 to 193.82 cm, with an average of 120.07 cm (Table 1). The average PL was 22.82 ± 4.60, ranging from 7.02 to 40.38 cm. For 1IN, the mean value of the entire population was 38.04 cm, ranging from 21.86 to 63.57 cm, with 93% for broad sense heritability ($h^{2}$). For 2IN, the mean value of the entire population was 23.30 cm, ranging from 11.02 to 35.74 cm, with 82% $h^{2}$. For 3IN, the mean value of the entire population was 17.76 cm, ranging from 6.27 to 29.14 cm, with 77% $h^{2}$. For 4IN, the mean value of the entire population was 12.81 cm, ranging from 2.42 to 27.16 cm, with 76% $h^{2}$. Similar results were obtained in 2019 (Table 1). These indicate that 4IN had the shortest length. With the exception of PL in 2018, all the traits evaluated had high $h^{2}$ in the range of 76–99%, pinpointing that the studied traits are largely under genetic control.

### 2.2. Genetic Diversity Revealed by SSR Molecular Markers

A total of 2664 alleles were detected by the 262 SSR markers across the 504 accessions, with an average ≈10.17 per locus (Appendix A). Allele number per locus ranged from 2 (RM7403, RM437 and RM7163 on chromosomes 3, 5 and 11, respectively) to 25 (RM7545 on chromosome 10) (Appendix A). The average genetic diversity of all SSR loci was 0.73, and the variation range per locus was 0.08 (RM7163 on chromosome 11) to 0.94 (RM7545 on chromosome 10) (Appendix A). The average PIC value was 0.70, ranging from 0.07 (RM7163 on chromosome 11) to 0.94 (RM7545 on chromosome 10), with a major distribution between 0.75 and 0.85 (Appendix A). The majority of SSR loci (86.26%) were highly informative (PIC > 0.5), while 10.69 and 3.05% of the loci were moderately and slightly informative (0.5 > PIC > 0.25 and PIC < 0.25), respectively, showing that this population possesses a high genetic diversity and genetic information; thus, it is suitable for GWAS.

### 2.3. Population Structure and Genetic Relatedness

A model-based population structure analysis revealed a significant population stratification among the 504 rice accessions. In this analysis, the log-likelihood values increased with the increase of subpopulation number (Appendix A); hence, the statistic ΔK as the diagnostic criterion was then used to determine a suitable number of subpopulation K, based on the method reported by Evanno, et al. [34]. The maximum ΔK value was observed at K = 8 compared to the other values of K; thus, the 504 accessions can be stratified by eight subpopulations (sub-pops) (Appendix A). Based on Q matrix information, each accession with Q value > 0.9 was assigned to the corresponding subpopulation (Appendix A). The 474 accessions were stratified into eight subpopulations, with 30 accessions being assigned into an admixture group. The neighbor-joining tree constructed based on the Nei’s genetic distances revealed that the 474 accessions were divided into eight clusters (Appendix A), which is consistent with the results from the STRUCTURE analysis. The grouping in model-based population and neighbor-joining tree followed geographical regions of the germplasm with minor discrepancies. For instance, Sub-Pop 1 and Cluster I comprised mainly germplasm from Eastern China, Sub-Pop 5 and Cluster V consisted with germplasm mainly from Northeast China, and Sub-Pop 6 and Cluster VI included mainly germplasm from Vietnam.

The genetic relatedness analysis in this study indicated that over 83% of the kinship coefficient values (K matrix) were in a range of 0–0.25 based on the 262 genetic SSR markers, and the remaining 16.8% showed various degrees of genetic relatedness, explaining the existence of a weak relatedness between pairwise accessions used (Appendix A). Furthermore, a K matrix was obtained through the results from genetic relatedness analysis for association analysis.

### 2.4. Genetic Differentiation among Subpopulations

The eight sub-pops obtained from the model-based approach were also subjected to analysis of molecular variance (AMOVA) to evaluate the percentage of variation among and within sub-pops. AMOVA showed that 48.21% of the total genetic variation occurred among the sub-pops, whereas the remaining 51.79% was attributed to the individual differences within sub-pops (Appendix A). Fixation index (*Fst*) statistics was used to test the genetic variation in the eight sub-pops. Pairwise *Fst* values indicated significant differentiation among all the pairs of sub-populations in a range of 0.41 (Sub-Pop 5 vs. Sub-Pop 3) to 0.59 (Sub-Pop 8 vs. Sub-Pop 2), suggesting that all the eight sub-pops were significantly different (*p* < 0.01) from each other. The corresponding standard Nei’s genetic distance between the eight sub-pops ranged from 0.48 to 0.75. Sub-Pops 2 and 8 were more differentiated from each other as per the *Fst* estimates (Appendix A). To sum up, the results of AMOVA and *Fst* analyses were consistent with the results generated through phylogenetic tree-based and STRUCTURE analyses. These highlight the presence of statistically moderate genetic diversity and high diverse population structure.

Appendix A shows the standardized disequilibrium coefficient ($D^{\prime }$) values of significant linkage disequilibrium (LD) pairwise loci in all of sub-pops calculated at *p* < 0.05. Among the 34,191 pairs, 9553 pairs indicated a presence of significant LD, including both inter- and intra-chromosomal combinations. The lowest percentage of significant LD pairwise loci was found in Sub-Pop 7 (3.08%), whereas the highest one was recorded in Sub-Pop 5 (8.83%), followed by Sub-Pop 3 (8.63%) (Appendix A). From the average of $D^{\prime }$ values, we see that Sub-Pop 4 had the highest $D^{\prime }$ (0.84), and Sub-Pop 6 had the lowest average of $D^{\prime }$ (0.63) among the eight sub-pops.

### 2.5. Phenotypic Variation of the Six Traits in the XC-CSSL Population and Its Parents

Table 2 shows descriptive statistics for each trait calculated from the 53 lines in 2018 and 2019. The two parents differed in PH: Xiushui79 (86.04 ± 0.54 and 87.08 ± 0.70 cm) compared with C-bao (94.48 ± 1.11 and 95.84 ± 1.58 cm) across the two years. The PH of the XC-CSSL ranged from 72.90 to 141.96 cm across the two years. There was no statistically significant difference between the two years for each trait, whether it was the parent population (Xiushui79, C-bao) or the segregating XC-CSSL population. A significantly longer PL was recorded in C-bao (25.32 cm and 26.38 cm) than in Xiushui79 (15.03 cm and 15.92 cm) across two years. The PL in XC-CSSL population ranged from 13.46 to 26.82 cm in the 2 years.

In 2018, for the internode traits, the XC-CSSL population had a mean length in the range of 28.54 ± 5.21 cm for 1IN, 18.07 ± 2.84 cm for 2IN, 15.97 ± 2.69 cm for 3IN and 9.12 ± 2.57 cm for 4IN. Compared to the parental lines, for Xiushui79 mean length was 24.81 ± 1.02 cm for 1IN, 17.30 ± 0.35 cm for 2IN, 16.30 ± 0.61 cm for 3IN and 7.04 ± 0.78 cm for 4IN; and for the C-bao parent, the order was 34.06 ± 0.64 cm for 1IN, 15.99 ± 0.26 cm for 2IN, 10.10 ± 0.27 cm for 3IN and 3.77 ± 0.82 cm for 4IN. Similar results were observed in 2019 (Table 2). The $h^{2}$ ranged from 71.39 and 96.97% for the six traits studied in the 2 years, suggesting that the variation in all the traits under study were largely under genetic control, rather than under the control of environment effects.

### 2.6. Association Mapping and Discovery of Favorable Alleles for PH and Its Component Traits in the Natural Population

All the accessions with Q < 0.9 were removed in the present study, and the remaining 474 accessions were utilized to conduct marker–trait association mapping. The MLM approach of marker–trait association was applied, using the Q + K model in TASSEL 5.0 with *p* < 0.05 detected a total of 109 markers associated with the six traits evaluated in this study across two years (Table 3 and Appendix A). Of these, 40 markers were significantly associated with the six traits in both 2018 and 2019 (Table 3), of which six markers were in regions where the QTL associated with PH or either of its component traits had been identified and reported in previous studies, whereas the remaining 69 markers were environment/year specific (Appendix A). The 40 markers were unevenly distributed across 11 out of the 12 chromosomes in rice.

In this study, the marker alleles with positive effect were considered favorable alleles for PL and 1IN, while those with negative effect were considered elite alleles for PH, 2IN, 3IN and 4IN. The larger the positive or negative effect allele value was, the better the allele for the trait measured. For the thirty-four new loci, 139 favorable alleles were identified for practical breeding purposes targeted at ideal plant height architecture.

#### 2.6.1. SSR Association Loci and Favorable Alleles for PH in the Natural Population

Five marker loci, i.e., RM301, RM348, RM305, RM264 and RM512, for PH were detected in the natural population on chromosomes 2, 4, 5, 8 and 12, respectively, in both 2018 and 2019 (Table 3). The PVE ranged from 2.41 to 6.92%. The highest PVE was observed on RM264, explaining 5.16% of phenotypic variance in 2018 and 6.92% in 2019. RM305 and RM264 loci for PH were also detected by Jiang, et al. [35] and Bai, et al. [36], respectively.

All the phenotypic effect values of the four favorable alleles (with negative phenotypic effect) at RM301 locus showed significant differences between each other except between RM301-150 bp and RM301-170 bp, as revealed by *p*-value and its significance of *F* test of ANOVA for the locus and the *t*-test method of protected least significant difference (PLSD) (Appendix A). The phenotypic effect values of the two favorable alleles at the RM348 locus showed significant differences, as revealed by the same test method used above (Appendix A). Among the 15 pairs of comparison of the phenotypic effect values constructed by the six favorable alleles at locus RM305, thirteen pairs of comparisons showed significant variation (Appendix A). The allele RM305-210 bp showed the largest phenotypic effect (−51.42 cm) for reducing PH, followed by a close proximity phenotypic effect of −43.31 cm being controlled by the favorable allele RM305-370 bp. In total, 210 of 474 accessions carried the allele RM305-210 bp, with its typical cultivar being Maoguangdao, and eight accessions carried the allele RM305-370bp, with Yue4 as a typical carrier (Appendix A). The allele RM264-150 bp showed the largest phenotypic effect (−33.02 cm) for reducing PH in this locus, with Nannongjing2R being its typical cultivar, and 21 accessions carried the allele RM264-150 bp. On the other hand, the allele RM512-220 bp had the highest phenotypic effect (−24.00 cm) in this locus for reducing PH, with its typical cultivar being Youmang429, and a total of 58 accessions carried the allele RM512-220 bp.

According to the phenotypic values and the number of favorable alleles that could be substituted or pyramided into an individual plant, the top five cross-combinations were predicted for PH, and their corresponding phenotypic increment effects (cm) are listed in Appendix A. When Cai1 is crossed with Nannongjing2R, all of the favorable alleles at the 5 loci could be combined into a single genotype, and this could lead to a reduction of length of −30.57 cm.

#### 2.6.2. SSR Association Loci and Favorable Alleles for PL in the Natural Population

For PL, nine significant associated loci were detected in the natural population in both years (Table 3). The loci were distributed on chromosomes 1, 3, 5, 6, 7, 9 and 10. RM3533 on chromosome 9 at 65.1 cM accounted for the highest PVE, 7.82%, in 2018 and 7.66% in 2019. RM1 and RM3600 loci for PL were also detected by Hong and Masahiko [37] and Liu, et al. [19], respectively.

All the phenotypic effect values of the thirty-four favorable alleles (with positive phenotypic effect), except two at the RM583 locus, showed significant differences between each other. The average highest positive phenotypic effect value was observed on RM3330-135 bp, RM5380-115 bp and RM3533-220 bp, with 7.44, 8.58 and 9.44 cm, respectively. The phenotypic effect values of the six favorable alleles at the RM3600 locus showed significant differences, as revealed by the *t*-test of PLSD (Appendix A). Among the six pairs of comparison of the phenotypic effect values constructed by the four favorable alleles at locus RM3533, four pairs of comparisons showed significant difference (Appendix A). The favorable allele RM3533-220 bp on chromosome 9 had the highest positive phenotypic effect (9.44 cm) for increasing this trait, and the typical carrier accession was Yue63 (Appendix A), while the highest negative phenotypic effect (−10.60 cm) was detected on favorable allele RM338-100 bp, occupying chromosome 3, with its typical carrier accession being Datougui (Appendix A).

#### 2.6.3. SSR Association Loci and Favorable Alleles for 1IN in the Natural Population

For 1IN, one marker each on chromosomes 2 (RM301), 8 (RM264) and 9 (RM257) was significantly associated in both years in the natural population (Table 3). Among the loci, RM257 on chromosome 9 had the largest PVE (5.94%) for the 2 years. For this trait, all the phenotypic effect values of the four favorable alleles (with positive phenotypic effect) at the RM301 locus showed significant differences between each other (Appendix A). The phenotypic effect values of the six favorable alleles at the RM264 locus showed significant differences, as revealed by the *t*-test (Appendix A). Among the 15 pairs for comparison of the phenotypic effect values constructed by the six favorable alleles at the locus RM257, fourteen pairs of comparisons showed significant differences (Appendix A). The maximum phenotypic positive effect for increasing this trait (10.42 cm) was observed on RM264-120 bp on chromosome 8, with Heizhong as the typical carrier, followed by RM257-125 bp (9.62 cm) on chromosome 9, carried by Huangkewanguangtou (Appendix A), and the average highest negative phenotypic effect value (−6.62 cm) was observed on RM301-150 bp, with Zacaodao as the typical carrier variety (Appendix A).

Appendix A summarizes the favorable alleles carried by the superior parents and corresponding phenotypic effect for PL and 1IN. Taking into account the phenotypic values and the number of favorable alleles that could be substituted or pyramided into an individual plant, we see that the parent Yue38 for PL possess all nine favorable alleles, and the parents Heizhong and Qiyunuo10 for 1IN possess all three favorable alleles at all their loci that could be combined into a single cultivar. Notwithstanding, the top five cross-combinations predicted to make a longer PL, and 1IN and its corresponding phenotypic effect (cm) are listed in Appendix A. For example, for Yue38 × Qiyunuo10, it was predicted that 12 favorable alleles could be pyramided into a single genotype, resulting in an improvement of 9.38 cm in both traits.

#### 2.6.4. SSR Association Loci and Favorable Alleles for 2IN in the Natural Population

For 2IN, six markers were observed in the natural population distributed on chromosomes 2, 4, 5 and 12 associated, and their PVE values ranged from 3.85 to 5.52%, of which RM301 on chromosome 2, found at 53.5 cM, had the highest PVE (5.52%) for both years.

All the phenotypic effect values of the nineteen favorable alleles (with negative phenotypic effect), except four and three at the RM301 and RM305 loci, respectively, showed significant differences between each other (Appendix A). All three pairs of comparisons of the phenotypic effect values constructed by the three favorable alleles at the RM348 locus revealed significant differences (Appendix A). The favorable allele RM348-130 bp on chromosome 4 showed the largest phenotypic effect (−7.48 cm) for reducing this trait, followed by RM512-115 bp (−5.55 cm) residing on chromosome 12, and both being carried by the typical carrier accession Zhonghua3 (Appendix A). The highest phenotypic positive effect value (11.33 cm) was found on RM348-170 bp (chromosome 4) with the typical carrier variety, Yue4 (Appendix A).

#### 2.6.5. SSR Association Loci and Favorable Alleles for 3IN in the Natural Population

For 3IN, eight significant association loci were detected in both years in the natural population across chromosomes 2, 4, 5, 7, 8 and 12, with the PVE ranging from 2.47 to 6.67%. RM264 on chromosome 8 had the highest PVE, 4.97% and 6.67%, in 2018 and 2019, respectively (Table 3).

All the phenotypic effect values of all the favorable alleles (with negative phenotypic effect) out of two alleles at the locus RM122 presented significant difference between each other (Appendix A). Among the 28 pairs of comparison of the phenotypic effect values constructed by the eight favorable alleles at locus RM3589, seventeen pairs of comparisons showed significant difference, as revealed by the *t*-test (Appendix A).

The identified thirty-five favorable alleles had an average negative phenotypic effect value ranging from −7.33 to −0.23 cm. Among them, the favorable allele RM247-165 bp on chromosome 12 displayed the largest phenotypic effect (−7.33cm) for reducing this trait, followed by RM264-150 bp (−6.62 cm) on chromosome 8, with Baikenuo and Nannongjing2R as typical carrier accessions, respectively (Appendix A). In contrast, the highest phenotypic positive effect value (9.86 cm) for the 3IN was observed on RM348-170 bp (chromosome 4), with Yue4 as the typical carrier variety (Appendix A).

#### 2.6.6. SSR Association Loci and Favorable Alleles for 4IN in the Natural Population

For 4IN, nine association markers were identified in the natural population for both years, distributed on chromosomes 2, 4, 5, 6, 7, 8 and 12 (Table 3). RM264 on chromosome 8, at 138.2 cM, explained the maximum phenotypic variation for 4IN, namely 6.39% in 2018 and 6.19% in 2019 (Table 3). Comparatively, three markers (RM512, RM348 and RM305) were co-associated with PH, 2IN, 3IN and 4IN (Table 3). Moreover, RM301 was associated with PH, 1IN, 2IN, 3IN and 4IN concurrently. Moreover, RM264 on chromosome 8 was linked to PH, 1IN, 3IN and 4IN, whereas RM122 on chromosome 5 was associated with PL, 2IN, 3IN and 4IN (Table 3). RM225 and RM512 for 4IN were also detected by Wang, et al. [38] and Yamamoto, et al. [39], respectively.

All the phenotypic effect values of the four favorable alleles (with negative phenotypic effect) at the RM301 locus showed significant differences between each other, except between the alleles RM301-145 and RM301-150 (Appendix A). The phenotypic effect values of the seven favorable alleles at the RM264 locus showed significant differences, as revealed by the *t*-test (Appendix A). In this trait, 21 pairs of comparison of the phenotypic effect values were constructed by the seven favorable alleles at the locus RM234, of which twelve pairs of comparisons revealed significant differences, as shown by the *t*-test (Appendix A).

The maximum phenotypic effect (−7.19 cm) for reducing this trait was observed on RM264-150 bp (chromosome 8), with Nannongjing2R as the typical carrier accession (Appendix A), and the highest positive phenotypic effect (14.15 cm) was detected on RM234-105 bp located on chromosome 7, carried by Buxienuo (Appendix A).

Fifteen excellent parental combinations were proposed for pyramiding favorable alleles into one single genotype to improve the rice basal internode length (Appendix A).

### 2.7. QTL Identification for PH and Its Component Traits in XC-CSSL Population

Table 4 shows the nine QTLs detected for the PH and its component traits in the XC-CSSL population, based on the RSTEP-LRT method in the CSL option in QTL IciMapping version 4.1 software, with an LOD threshold of 2.5 over two years.

For the PH, one QTL (*qPH-12*) linked to marker RM19 was detected on chromosome 12, with PVE 17.77% in 2018 and 17.19% in 2019. The additive effect (or substitutive effect) of allele from C-bao was 15.10 cm in 2018 and 14.85 cm in 2019 (Table 4). The allele from parent Xiushui 79 at *qPH-12* locus was the favorable allele in this study. Interestingly, the locus *qPH-12* is the same as the one linked to the locus RM512 in the natural population for PH (Table 3 and Table 4).

For PL, one QTL (*qPL-7*) linked to RM180 was identified in 2018, with a PVE of 22.95% and additive effect of 2.28 cm, and the beneficial allele at this locus was inherited from C-bao.

For 1IN, one QTL (*q1IN-7*) linked to marker RM180 was identified on the same chromosome region of *qPL-7*, with the largest PVE of 28.42% and an additive effect of 5.31 cm of alleles from C-bao. The QTL *q1IN-9* linked to marker RM257 had a PVE of 16.92%, with an additive effect of 5.68 cm.

For 2IN, three QTLs (*q2IN-7.1*, *q2IN-7.2* and *q2IN-12*) linked with RM180, RM542 and RM19, respectively, were detected across two years, where QTL *q2IN-12* showed the highest PVE of 10.78%, with a maximum additive effect of 2.87 cm of allele from C-bao.

For 3IN, one QTL (*q3IN-12*) was detected linked with RM19, with a PVE of 16.70% and additive effect of 2.82 cm. The allele from parent C-bao at *q3IN-12* locus was the favorable allele in this study.

For 4IN, the QTL *q4IN-8* linked to RM72 was identified in 2018, with a PVE of 14.15%, and the beneficial allele at this locus came from the C-bao genome.

### 2.8. Comparison of Mapping Results in the Natural and XC-CSSL Populations

The marker locus RM257 (48.8 cM) located on chromosome 9, which was linked with the 1IN, was detected via both association and linkage mapping strategies (Table 3 and Table 4). In the XC-CSSL population, RM257-125 bp was the favorable allele derived from the C-bao parent. In the natural population, RM257-125 bp was also the favorable allele and was carried by Huangkewanguangtou as the typical carrier variety.

## 3. Discussion

In the natural population, we detected five SSR markers associated with PH in two years. Among them, the marker RM305 on chromosome 5 was consistent with the QTL *qPH5* detected in previous studies linked between the marker interval RM164-RM305, with an average explained phenotypic variation of 3.5% in recombinant inbred lines [35]. Similarly, Bai, et al. [36] reported one major QTL, *qph8*, located on chromosome 8 within the marker interval RM502–RM264 with a logarithms of odd (LOD) score more than 10 and explained over 24% of the phenotype variance among the recombinant inbred lines. The remaining three QTLs linked by RM301, RM348 and RM512 were newly found in the present study, and they could be targeted for future studies to broaden our understanding of PH in rice. After pyramiding the favorable alleles located on these five loci into a single genotype, the PH could be reduced by −30.57 cm. A moderate plant height is an important basis for rice breeding, and this is supported by Andrew-Peter-Leon, et al. [40] and Bhuvaneswari, et al. [41], who have reported that *OsGA20ox2* and *OsFBX26*, have important roles in regulating semi-dwarfism and early flowering in rice. Likewise, this finding is supported by Liu, et al. [29], who reported that several semi-dwarf rice varieties have been bred by genetic introduction of the mutant alleles of SD1, which encodes OsGA20ox2 for catalyzing GA biosynthesis. This is true, since other studies have shown that *TUT1* encodes an inhibitor of the cAMP-like receptor protein, and *es1-1* displays a shorter plant height and shorter internode [42]. Considering that *es1-1* affects both plant height and internode length, it is reasonable to presume that a desirable plant height could be achieved by *es1-1* regulation. Interestingly, *es1-1* is also important for actin assembly and panicle development [22].

For PL, nine SSR associated markers were detected in both years, suggesting that these markers were stable against environmental factors. RM3600 is a stable marker which was found in other studies associated with PL [19], and the remaining eight SSR markers detected for this trait, namely RM1, RM583, RM338, RM122, RM3330, RM5380, RM3533 and RM6160, are reported here for the first time and represent promising targets for further analysis of PL. In keeping with previous studies, we found that RM3600 on chromosome 9 was consistent with the QTL *LP1* delimited to a 1.60 cM region between RM3600 and RM410, detected via both association mapping and linkage analysis, and has been identified as a major QTL associated with high phenotypic variation for PL; and the *LOC_Os09g28300* was validated as the gene locus controlling PL [19]. In this trait, the elite allele RM3533-220 bp had the largest phenotypic effect value among all positive elite alleles found on their marker loci (Appendix A), suggesting that this elite allele may be valuable for practical breeding to increase the PL and, in turn, the rice yield, due to the strong effects of the PL on yield components, such as grain number per panicle, and therefore can be used as candidate selection criteria for yield breeding [19]. After pyramiding the favorable alleles from the nine loci into a single genotype, the PL could increase by 4.48 cm (Appendix A).

For 1IN, all three QTLs detected in the natural population were novel. We found that the allele RM264-120 bp had the largest PEV among the six elite alleles found at this marker locus (Appendix A). This indicates that this favorable allele could be used to increase the length of the 1IN in breeding programs and, consequently, would be of great value for panicle exertion enhancement. It is supported by previous studies that the elongation of different internodes in the rice culm may be regulated by different genes, and the loss of function or downregulation of *EUI1*, which encodes a P450, *CYP714D1*, gave rise to the increase of 1IN [24,25,26]. After pyramiding the favorable alleles from the three loci into a single genotype, the 1IN length might increase to 3.08 cm (Appendix A).

For 2IN, the six QTLs for this trait in the natural population were newly detected, and the favorable allele RM348-130 bp on chromosome 4 exhibited pleiotropic effects on reducing the PH and the basal internode lengths (Table 3). Moreover, since all three pairs of comparisons of the phenotypic effect values constructed by the three favorable alleles at RM348 locus revealed significant difference (Appendix A), we speculate that this favorable allele is an important candidate and could be further validated trough marker-assisted selection (MAS) and that the genes on this locus could be mapped and functionally validated to deepen our understanding on genetic bases of PH, 2IN, 3IN and 4IN, which could be valuable for breeding programs designed to increase lodging resistance based on selecting desirable cultivars in rice. After pyramiding the favorable alleles from the six loci into a single genotype, the 2IN length is likely to be reduced by −2.53 cm. Though studies on 2IN development in rice remain limited, Jiang, et al. [43] reported a gene, *SSI2*, responsible for reduced elongation specifically in the second internode counted from the top, which encodes a fatty-acid dehydrogenase and has an effect on the process of fatty-acid biosynthesis.

For 3IN, all the eight QTLs were detected for the first time in the natural population. RM264 on chromosome 8 showed the largest PVE for the 3IN and 4IN traits (Table 3), suggesting that this marker can be validated and used for MAS in developing lodging resistance of the basal internodes in rice. After pyramiding the favorable alleles from the eight loci into a single genotype, the 3IN length possibly can be reduced by −2.29 cm. For 4IN, nine QTLs were detected in the natural population, of which RM225 and RM512 for 4IN were also detected by Wang, et al. [38] and Yamamoto, et al. [39], respectively, and the remaining seven marker loci, namely RM301, RM348, RM122, RM305, RM598, RM234 and RM264, were novel. Previous studies by Wang, et al. [38] reported a QTL significantly associated with the fourth internode, *qFOIL-6-4*, tagged by RM225, with an average explained phenotypic variation of more than 3.78% in 331 japonica rice accessions. In addition, RM301, RM122 and RM512 loci on chromosomes 2, 5 and 12, respectively, had pleiotropic effects on reducing the 2IN, 3IN and 4IN, suggesting that these loci had the most effective control on internode elongation. It is reported that the expression of the QTL flanked by RG20q and RG901 on chromosome 12 stimulated panicle initiation at the early stage, peaking at approximately 7 days later, and then declining gradually, with effects on the 2IN, 3IN and 4IN [23].

In the present study, we also found that the favorable alleles RM247-165 bp and RM264-150 bp (Appendix A) exerted the maximum phenotypic effect for reducing the 3IN and 4IN. This is supported by previous studies which reported that the phenomenon of lodging in rice often occurs at lower internodes (below the 3IN from the top) in response to bending higher up the stem; therefore, the lower internodes tend to result in better lodging resistance [44], and this is consistent with other previous reports on rice [45] and wheat [46] that supported this idea, where they found that PH markers, especially those affecting the two lower internodes (3IN and 4IN) are expected to be able to contribute significantly to lodging resistance. Previous studies in rice plants indicate that the reduced height of the *sd-1* is due to the defect in the GA biosynthetic pathway [6,47,48], as opposed to the wheat Rht-B1/Rht-D1mutant plants in which the GA signal transduction pathway is weakened [49,50]. Importantly, the OsGA20ox gene corresponds to the *Sd-1* locus in the rice genome [41,51]. Similar results were found by Liu, et al. [29], who reported that the culm content of bioactive gibberellin, especially in the basal internodes, is reduced by a semi-dominant lodging resistance gene, *SBI*, which encodes a culm-specific expressed *OsGA2ox*, and, thereafter, the higher activity of the *SBI* considerably reduced the rice PH and contributed to breed enhanced lodging resistant rice varieties and, therefore, increased the yield. For this reason, the shorter the 4IN, 3IN and 2IN lengths are, the better for no lodging in rice.

Our analysis of the XC-CSSL population revealed that the PH would be shortened 29.8 cm if the chromosome region of the QTL *qPH-12* linked to RM19 in C-bao was substituted by Xiushui 79 (Table 4); thus, the lodging resistance can be improved in rice varieties, and this is supported by Hirano, et al. [52], who reported that reducing the PH lowers the center of gravity and improves bending-type lodging resistance. For the PL trait, the allele from C-bao on QTL *qPL-7*, a newly discovered locus linked with the marker RM180 with PVE 22.95%, could increase PL 2.28 cm (Table 4). This newly detected locus needs further screening and validation to possibly be used directly for improving the PL of the japonica rice variety by MAS. For 1IN, two QTLs (*q1IN-7* and *q1IN-9*) were detected across 2 years. The QTL *q1IN-7* had the highest PVE 28.42% observed among all loci, and the largest addictive effect value, i.e., 5.68 cm, was found on *q1IN-7* closely linked with the marker RM257, and, interestingly, this marker locus was also detected in the natural population. In this study, the RM257-125 marker allele was the favorable allele in both populations. Although the method and detection principle of the XC-CSSL and natural populations for mapping QTLs differed, the RM257-125 bp can be detected consistently, showing that the allele RM257-125 bp is stable and has a strong effect on the 1IN. The alleles increasing the first internode length in these two loci (*q1IN-7* and *q1IN-9*) were from the C-bao parent. Notwithstanding, the first internode length will increase by 10.99 cm since the chromosome region of the two loci are from C-bao genome. For 2IN, three QTLs (*q2IN-7.1*, *q2IN-7.2* and *q2IN-12*) were detected across 2 years. The QTL *q2IN-12* had the highest PVE 10.78% observed among all new loci, and the largest addictive effect value, i.e., 2.87 cm, was found in this locus linked with the marker RM19. The alleles reducing the second internode length in these three loci were from the C-bao parent. Thus, the second internode length of Xiushui 79 would decrease by 7.74 cm if the chromosome regions of the two loci are substituted by the C-bao region. For 3IN, one QTL, *q3IN-12*, was detected with a PVE of 16.7% and an addictive effect value of 2.82 cm. If the chromosome region of this site in C-bao is substituted by Xiushui 79, the third internode length will be shortened 5.64 cm. For 4IN, one QTL, *q4IN-8*, was detected with a PVE of 14.15% and additive effect value of 1.40 cm. The 4IN will be shortened by 2.80 cm if the chromosome region of this site in C-bao is substituted by Xiushui 79 genome.

These new loci provided in this study serve as attractive candidate regions for further study on elucidation of the genetic basis for the improvement of rice-plant height architecture and its component traits through MAS.

## 4. Materials and Methods

### 4.1. Plant Materials

Two mapping populations, namely a natural population and a chromosome segment substitution line (CSSL) population, were used in the present study. The natural population consisted of 504 rice accessions from different geographical regions (Appendix A). The seeds of all accessions were obtained from State Key Laboratory of Crop Genetics and Germplasm Enhancement, Nanjing Agricultural University, Nanjing, China. The CSSL population was composed of 53 lines derived from a cross between two japonica rice cultivars, Xiushui79 (background parent, Figure 2A, left side) and C-bao cultivar (donor parent, Figure 2A, right side). This XC-CSSL population construction has previously been completed and published by Bux, et al. [53].

### 4.2. Field Planting and Trait Measurement

All the genetic materials were planted in randomized complete block design with three replications in the paddy fields at Jiangpu Agricultural Experiment Station of Nanjing Agricultural University, Nanjing, China (32°05′ N, 118°64′ E), over two seasons, 2018 and 2019. The seeds of both populations were sown in the seedling nursery bed of paddy fields on 10 May and 16 May in 2018 and 2019, respectively. The healthy and uniform seedlings were transplanted at approximately 30 days after sowing on nursery beds onto the field at a spacing of 17 × 20 cm with one seedling per hill. Each plot consisted of four rows with eight hills per row. The fields were managed following local standard agronomic practices. At the mature stage, three plants in the middle rows of each plot from each replication were randomly sampled for measurements of the six traits, i.e., PH, PL, 1IN, 2IN, 3IN and 4IN of the tallest stem within a hill (Figure 1B). PH was recorded from the ground surface to the tip of the tallest panicle, excluding awns. PL was recorded as the length from the panicle neck node to the panicle tip of the tallest panicle. All lengths were recorded in centimeters. The means of three replicates were used in the data analysis.

### 4.3. Phenotypic Data Analysis and Heritability in a Broad Sense

The phenotypic data collected on PH and its associated traits were subjected to analysis of variance (ANOVA) based on the mean values for environment, using SAS statistical software version 9.4 (SAS Institute Inc., Cary, NC, USA). Broad-sense heritability ($h^{2}$) was calculated for each environment and population, using the following formula:(1)$$h^{2}=\sigma_{g}^{2}/\left(\sigma_{g}^{2}+\sigma_{e}^{2}/n\right)$$
where $\sigma_{g}^{2}$ represents the genetic variance, $\sigma_{e}^{2}$ represents the error variance and *n* represents the number of replicates.

### 4.4. DNA Extraction and SSR Marker Genotyping

The methods described by Murray and Thompson [54] were applied to extract genomic DNA from young leaf tissue of each accession. A total of 262 simple sequence repeat (SSR) markers previously published on rice molecular map and microsatellite database [55,56] selected from the 12 rice chromosomes were used to genotype the natural population. The primers were synthesized by Shanghai Generay Biotech Co., Ltd., Shanghai, China. The PCR amplification was performed in a 10 µL reaction containing 1 µL of 20 ng µL^−1^ template DNA, 0.6 µL of 25 mmol L^−1^ MgCl_2_, 0.7 µL of 2 pmol µL^−1^ forward primers, 0.7 µL of 2 pmol µL^−1^ reverse primers, 0.2 µL of 2.5 mmol L^−1^ dNTP, 1 µL of 10 × PCR buffer, 0.1µL of 5 U µL^−1^ rTaq DNA polymerase (TaKaRa Bio Inc., Kusatsu, Shiga, Japan) and 5.7 µL of ddH2O. The DNA amplification was conducted by using a PTC-100, a new type Peltier Thermal Cycler (MJ Research Inc., Union, NJ, USA). The order of the PCR program in timing during the amplification was denaturation at 95 °C for 5 min, followed by 35 cycles of 95 °C for 30 s, 55 °C for 30 s, 72 °C for 30 s and a final extension step at 72 °C for 5 min. The amplified products were run on 8% non-denaturing polyacrylamide gels at a 180 V for approximately 100 min, and visualized through silver staining [57].

### 4.5. Population Structure Analysis, Genetic Diversity and Phylogenesis

The population structure (Q matrix) of 504 accessions was analyzed by using STRUCTURE software, version 2.2 [58], following the settings used by Sowadan, et al. [10]. Furthermore, we used an ad hoc statistic (ΔK) to determine the suitable K values [34]. All non-admixed individuals in each genetic subgroup were determined by using a Q value greater than 0.9 (Q > 0.9) in the Q-matrix. The PowerMarker on its version 3.25 was applied to carry out statistics, including the number of alleles per locus, major allele frequency, gene diversity and polymorphism information content (PIC) values per locus. Nei’s genetic distance [59] was calculated and used for the reconstruction of the phylogenic tree, using the neighbor-joining method, as implemented in PowerMarker, with the tree viewed by using MEGA X software [60].

Analysis of molecular variance (AMOVA) was conducted to assess population differentiation clustered by STRUCTURE 2.2, using the software Arlequin 3.5 [61]. Based on methods by Weir and Hill [62], the coefficient of subpopulation genetic fixation (*Fst*) was estimated among subpopulations, using Arlequin 3.5.

### 4.6. Linkage Disequilibrium Analysis

Linkage disequilibrium (LD) is the basis of association analysis, which refers to the non-random association of alleles of different loci in a population [63]. It includes both inter-chromosomal and intra-chromosomal combinations. TASSEL version 5 standalone software was used to compute the LD value between linked loci [64], using 100,000 permutations on all accession and on clusters derived by STRUCTURE 2.2. The standardized disequilibrium coefficients (D′) generally ranges from 0 to 1, and a D′ < 0.5 is a sign of LD decay. Before the association analysis, the rare alleles (with a frequency of ≤5%) were excluded from the dataset.

### 4.7. Association Mapping

Association mapping in the natural population was conducted with a mixed linear model (MLM) in TASSEL 5.0 software [65]. To avoid possible spurious associations between molecular marker and genomic locus, the K (the kinship matrix) and Q matrices were used as covariates in the association analysis [66]. The false discovery rate (FDR) of 0.05 was used as a threshold to control significant associations based on correction method reported by Benjamini and Hochberg [67]. Applying the association locus previously identified, the “null allele” (non-amplified allele) was used to calculate the phenotypic effect value of other alleles [68].

The formula used for calculating phenotypic effect value of a single allele was as follows:(2)$$a_{i}=\overset{n_{i}}{\underset{j=1}{\sum }}x_{ij}/n-\overset{n_{k}}{\underset{k=1}{\sum }}N_{k}/n_{k}$$
where *a_i_* was the phenotypic effect value of the allele of *i*, *x_ij_* indicates the phenotypic measurement values of *j* variety carrying the allele of *i*, *n_i_* denotes the number of materials carrying the allele of *i*, *N_k_* depicts the phenotypic value of the variety of *k* carrying the null allele and *n_k_* represents the number of materials carrying the null allele. Alleles with positive effect values are considered as an elite allele if the objective is to increase the trait value of interest; otherwise, if the objective is to reduce the trait, alleles of negative effect are considered as elite alleles, and we determine the carrier accessions accordingly. The alleles effect was subjected to ANOVA, and means separation was performed by using Duncan multiple range test.

### 4.8. QTL Detection for PH and Its Component Traits in the XC-CSSL Population

The Phenotypic data for PH and its component traits in the XC-CSSL population were analyzed based on the mean values for environment, using SAS statistical software version 9.4 (SAS Institute Inc., Cary, NC, USA). Since many lines of the XC-CSSL population comprised more than one segment from the donor parent, the standard *t*-test method is not appropriate for QTL detection. Therefore, we used the RSTEP-LRT mapping method proposed by Wang, et al. [69], which is equivalent to the standard *t*-test with the idealized CSS lines. The threshold of 2.5 was set, and the QTL detection was implemented in QTL IciMapping 4.1 software [69]. Furthermore, the QTL nomenclature was based on the principle reported by McCouch [70], but with minor modification.

## 5. Conclusions

There was high genetic variation in the natural population, due to the highly significant differences among the 504 accessions for plant height and its component traits. From the results that 34 stable novel loci for PH (3), PL (7), 1IN (3), 2IN (6), 3IN (8) and 4IN (7) were identified in the natural population, and the fact that *q1IN-7* linked with RM257 was also detected in the XC-CSSL population, we can conclude that there are many genetic loci controlling plant height and its components, and it is possible to shape plant height architecture by pyramiding various favorable alleles distributed at different loci in rice. Plant height in rice could be improved according to the crosses between the best parental combinations predicted for PL + 1IN (Yue38 × Qiyunuo10) and 2IN + 3IN + 4IN (Aijiaoluganhuang × Si4049 × Yandao6) in order to resolve the prevailing contradiction between increasing plant height and lodging.

## Figures and Tables

**Figure 1 plants-11-00999-f001:**
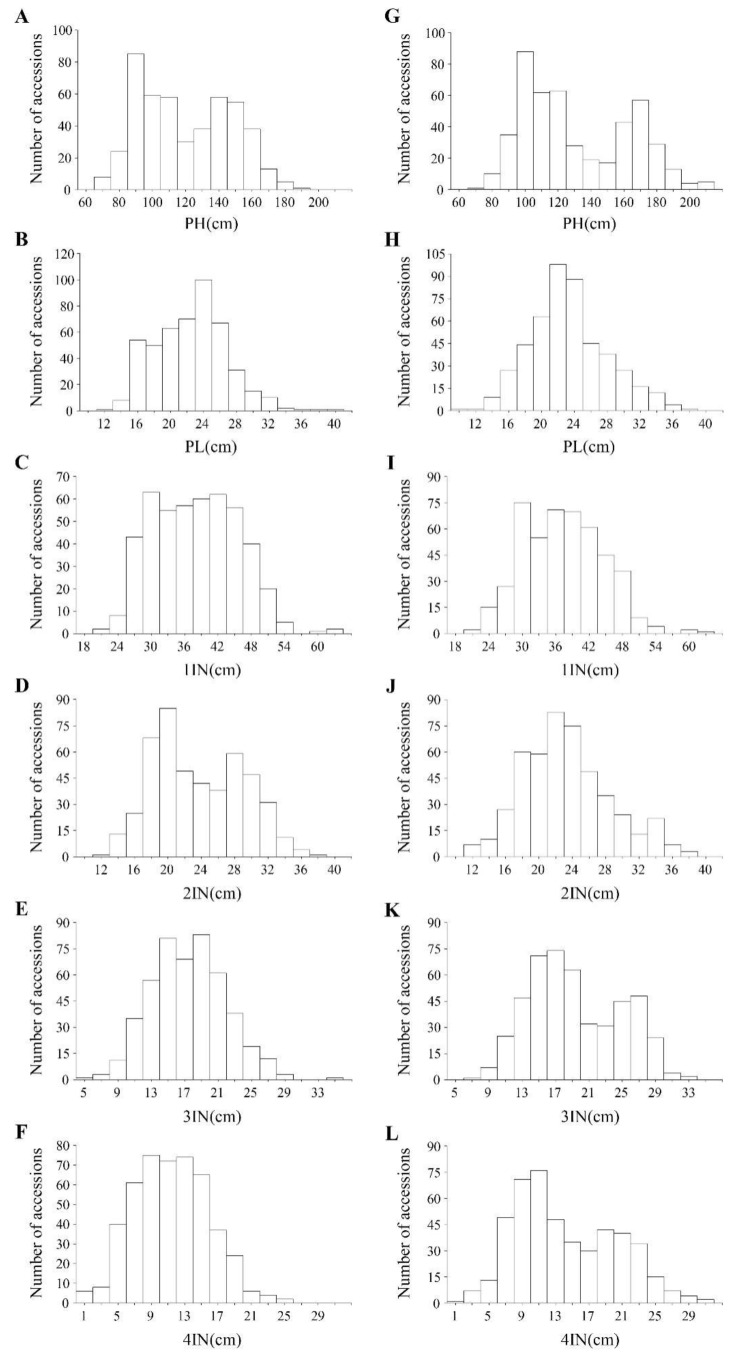
Frequency distribution of plant height and its component traits in the natural population in 2018 (**A**–**F**) and in 2019 (**G**–**L**). PH, plant height; PL, panicle length; 1IN, first internode from the top; 2IN, second internode from the top; 3IN, third internode from the top; and 4IN, fourth internode from the top.

**Figure 2 plants-11-00999-f002:**
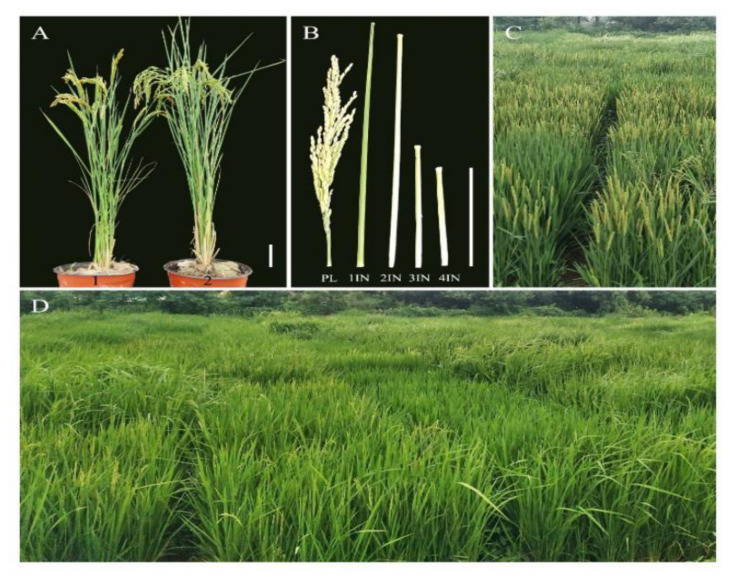
Plants, plant-height traits measurement and partial materials in XC-CSSL population and natural population. (**A**) Plants of *Xiushui79* (1) and *C-bao* (2) (scale bar = 5 cm). (**B**) Measurement of Panicle (PL) and internode length: the upper first (1IN), second (2IN), third (3IN) and fourth (4IN) internodes (Scale bar = 5 cm). (**C**) Partial materials of XC-CSSL population showing differences of plant height among lines. (**D**) Partial varieties of natural population showing differences in the plant height.

**Table 1 plants-11-00999-t001:** Descriptive statistics of phenotypic performance of 504 rice accessions for PH and its component traits in natural population across 2 years.

Trait (cm)	Year	Mean ± SD	Min	Max	Skew	Kurtosis	CV (%)	$h^{2}\;(\%)$
PH	2018	120.07 ± 27.24	68.78	193.82	0.22	−1.13	22.68	92.59
	2019	128.78 ± 29.90	75.73	230.27	0.42	−0.93	23.21	99.55
PL	2018	22.82 ± 4.60	7.02	40.38	0.35	0.45	20.16	45.27
	2019	23.32 ± 4.66	9.87	37.16	0.38	0.01	19.99	94.01
1IN	2018	38.04 ± 7.51	21.86	63.57	0.21	−0.57	19.75	93.00
	2019	37.30 ± 7.21	12.47	62.28	0.17	−0.05	19.32	95.03
2IN	2018	23.30 ± 5.10	11.02	35.74	0.30	−0.80	21.91	82.27
	2019	23.33 ± 5.22	11.48	37.63	0.45	−0.07	22.36	92.33
3IN	2018	17.76 ± 4.32	6.27	29.14	0.06	−0.52	24.33	77.20
	2019	19.30 ± 5.45	6.82	32.93	0.28	−0.87	28.26	94.55
4IN	2018	12.81 ± 5.37	2.42	27.16	0.45	−0.49	41.88	76.31
	2019	14.19 ± 6.03	1.96	31.44	0.41	−0.75	42.49	93.27

PH, plant height; PL, panicle length; 1IN, first internode from the top; 2IN, second internode from the top; 3IN, third internode from the top; 4IN, fourth internode from the top; SD, standard deviation; CV, coefficient of variation; $h^{2}$, heritability in broad sense.

**Table 2 plants-11-00999-t002:** Descriptive statistics of phenotypic variations in XC-CSSL population and its parents in two years.

Trait (cm)	Year	Xiushui79	C-bao	XC-CSSL Population						
		Mean ± SD	Mean ± SD	Mean ± SD	Min	Max	Skew	Kurtosis	CV (%)	$h^{2}\;(\%)$
PH	2018	86.04 ± 0.54	94.48 ± 1.11	90.64 ± 12.19	72.91	123.73	0.93	0.43	13.45	96.97
	2019	87.08 ± 0.70	95.84 ± 1.58	108.68 ± 12.79	86.2	141.96	1.08	0.67	11.76	94.93
PL	2018	15.03 ± 0.74	25.32 ± 0.57	16.91 ± 2.56	13.46	23.72	1.1	0.66	15.14	83.75
	2019	15.92 ± 0.63	26.38 ± 0.41	18.37 ± 2.62	15.5	26.82	1.35	1.18	14.25	89.71
1IN	2018	24.81 ± 1.02	34.06 ± 0.64	28.54 ± 5.21	22.72	44.64	1.55	1.48	18.26	93.95
	2019	26.43 ± 0.18	36.28 ± 0.73	29.20 ± 5.29	23.76	45.36	1.72	1.96	18.13	89.22
2IN	2018	17.30 ± 0.35	15.99 ± 0.26	18.07 ± 2.84	12.79	24.78	0.43	−0.2	15.71	94.92
	2019	18.31 ± 1.07	18.01 ± 0.80	21.99 ± 2.44	18.04	30.12	1.27	2.65	11.11	71.39
3IN	2018	16.30 ± 0.61	10.10 ± 0.27	15.97 ± 2.69	8.78	20.76	−0.49	−0.15	16.86	91.66
	2019	17.5 ± 0.06	12.23 ± 0.98	19.22 ± 2.41	12.94	26.14	−0.11	0.76	12.55	89.61
4IN	2018	7.04 ± 0.78	3.77 ± 0.82	9.12 ± 2.57	4.32	15.13	0.23	−0.31	28.2	79.83
	2019	8.40 ± 0.32	6.61 ± 0.76	12.87 ± 2.76	5.88	19.32	−0.22	0.46	21.49	81.09

PH, plant height; PL, panicle length; 1IN, first internode from the top; 2IN, second internode from the top; 3IN, third internode from the top; 4IN, fourth internode from the top; SD, standard deviation; CV, coefficient of variation; $h^{2}$, heritability in broad sense.

**Table 3 plants-11-00999-t003:** Marker–trait associations with *p*-value < 0.05, their equivalent false discovery rate probability (FDR), proportion of phenotypic variance explained (PVE) and marker position on chromosome derived from 262 markers and 474 rice accessions detected in two years.

Traits	SSR	Chr	Position	Position (bp) ^b^		2018			2019			Average	QTL Reported in the Previous Studies
	Marker		(cM) ^a^	Start Pos	End Pos	*p*-Value	PVE (%)	FDR	*p*-Value	PVE (%)	FDR	PVE (%)	References
PH	RM301	2	53.5	12,216,463	12,216,613	1.38 × 10^−3^	5.46	6.25 × 10^−3^	2.29 × 10^−3^	5.17	5.88 × 10^−3^	5.31	
	**RM348**	4	160.8	32,650,358	32,650,527	4.55 × 10^−2^	2.41	5.00 × 10^−2^	8.04 × 10^−3^	3.34	1.47 × 10^−2^	2.88	
	**RM305**	5	96.9	12,216,463	12,216,613	4.80 × 10^−3^	4.38	9.38 × 10^−3^	1.70 × 10^−2^	3.f65	1.76 × 10^−2^	4.01	[35]
	*RM264*	8	138.2	27,926,633	27,926,652	2.00 × 10^−2^	5.16	2.81 × 10^−2^	1.38 × 10^−3^	6.92	2.94 × 10^−3^	6.04	[36]
	**RM512**	12	39.4	5,104,270	5,104,483	1.55 × 10^−2^	3.00	1.25 × 10^−2^	2.72 × 10^−3^	3.90	8.82 × 10^−3^	3.45	
PL	RM1	1	25.4	4,635,793	4,635,870	3.30 × 10^−2^	4.50	4.71 × 10^−2^	8.97 × 10^−3^	5.39	1.76 × 10^−2^	4.94	[37]
	RM583	1	38.8	8,328,958	8,329,017	6.29 × 10^−3^	3.86	1.47 × 10^−2^	4.12 × 10^−3^	4.09	1.18 × 10^−2^	3.97	
	RM338	3	61.9	13,221,482	13,221,665	3.17 × 10^−2^	3.60	4.12 × 10^−2^	2.92 × 10^−2^	3.66	3.82 × 10^−2^	3.63	
	RM122	5	3	311,055	311,218	1.17 × 10^−2^	3.15	2.06 × 10^−2^	4.52 × 10^−4^	4.82	2.94 × 10^−3^	3.98	
	RM3330	6	61.6	11,064,125	11,064,294	5.54 × 10^−3^	6.69	8.82 × 10^−3^	4.39 × 10^−3^	6.85	1.47 × 10^−2^	6.77	
	RM5380	7	67	19,361,277	19,361,383	3.54 × 10^−3^	5.64	5.88 × 10^−3^	7.54 × 10^−4^	6.57	5.88 × 10^−3^	6.10	
	RM3533	9	65.1	17,833,841	17,833,864	1.03 × 10^−3^	7.82	2.94 × 10^−3^	1.31 × 10^−3^	7.66	8.82 × 10^−3^	7.74	
	RM3600	9	62.7	17,054,142	17,054,167	2.72 × 10^−2^	4.94	2.94 × 10^−2^	2.66 × 10^−2^	4.96	2.94 × 10^−2^	4.95	[19]
	RM6160	10	81	22,048,086	22,048,115	1.20 × 10^−2^	4.20	2.35 × 10^−2^	2.81 × 10^−2^	3.68	3.53 × 10^−2^	3.94	
1IN	RM301	2	53.5	12,216,463	12,216,613	7.99 × 10^−3^	4.44	8.33 × 10^−3^	1.70 × 10^−2^	3.98	3.00 × 10^−2^	4.21	
	*RM264*	8	138.2	27,926,633	27,926,652	3.11 × 10^−2^	4.84	4.17 × 10^−2^	2.27 × 10^−2^	5.06	3.50 × 10^−2^	4.95	
	RM257	9	48.8	17,570,591	17,570,612	2.20 × 10^−2^	3.83	3.33 × 10^−2^	5.76 × 10^−4^	5.94	5.00 × 10^−3^	4.88	
2IN	RM301	2	53.5	12,216,463	12,216,613	1.24 × 10^−3^	5.52	1.36 × 10^−2^	2.13 × 10^−2^	3.85	3.18 × 10^−2^	4.68	
	RM5427	2	84.6	21,544,386	21,544,417	1.18 × 10^−2^	5.20	3.18 × 10^−2^	1.80 × 10^−2^	4.92	2.73 × 10^−2^	5.06	
	**RM348**	4	160.8	32,650,358	32,650,527	7.39 × 10^−4^	4.57	4.55 × 10^−3^	1.29 × 10^−3^	4.29	1.82 × 10^−2^	4.43	
	RM122	5	3	311,055	311,218	1.87 × 10^−3^	4.10	1.82 × 10^−2^	4.71 × 10^−4^	4.80	9.09 × 10^−3^	4.45	
	**RM305**	5	96.9	20,944,257	20,944,466	1.89 × 10^−3^	4.89	2.27 × 10^−2^	1.21 × 10^−3^	5.14	1.36 × 10^−2^	5.02	
	**RM512**	12	39.4	5,104,270	5,104,483	9.03 × 10^−4^	4.47	9.09 × 10^−3^	4.67 × 10^−4^	4.80	4.55 × 10^−3^	4.63	
3IN	RM301	2	53.5	12,216,463	12,216,613	7.60 × 10^−3^	4.47	1.79 × 10^−2^	3.79 × 10^−3^	4.88	1.15 × 10^−2^	4.68	
	**RM348**	4	160.8	32,650,358	32,650,527	1.46 × 10^−2^	3.03	2.50 × 10^−2^	4.12 × 10^−2^	2.47	4.23 × 10^−2^	2.75	
	RM122	5	3	311,055	311,218	7.38 × 10^−4^	4.57	3.57 × 10^−3^	1.25 × 10^−3^	4.30	3.85 × 10^−3^	4.44	
	**RM305**	5	96.9	20,944,257	20,944,466	5.78 × 10^−3^	4.27	1.43 × 10^−2^	1.50 × 10^−2^	3.72	1.92 × 10^−2^	4.00	
	RM3589	7	89.8	25,054,610	25,054,631	2.83 × 10^−2^	5.21	4.64 × 10^−2^	3.16 × 10^−2^	5.13	3.46 × 10^−2^	5.17	
	*RM264*	8	138.2	27,926,633	27,926,652	2.63 × 10^−2^	4.97	4.29 × 10^−2^	2.08 × 10^−3^	6.67	7.69 × 10^−3^	5.82	
	RM247	12	26.7	3,185,544	3,185,599	1.85 × 10^−2^	4.90	2.86 × 10^−2^	4.71 × 10^−2^	4.25	5.00 × 10^−2^	4.58	
	RM512	12	39.4	5,104,270	5,104,483	4.51 × 10^−3^	3.65	1.07 × 10^−2^	5.05 × 10^−3^	3.59	1.54 × 10^−2^	3.62	
4IN	RM301	2	53.5	12,216,463	12,216,613	5.32 × 10^−4^	5.99	2.94 × 10^−3^	3.49 × 10^−3^	4.93	6.67 × 10^−3^	5.46	
	RM348	4	160.8	32,650,358	32,650,527	2.57 × 10^−2^	2.72	3.82 × 10^−2^	8.00 × 10^−3^	3.35	1.33 × 10^−2^	3.03	
	RM122	5	3	311,055	311,218	9.96 × 10^−4^	4.42	5.88 × 10^−3^	1.85 × 10^−3^	4.10	3.33 × 10^−3^	4.26	
	RM305	5	96.9	20,944,257	20,944,466	4.77 × 10^−3^	4.38	1.76 × 10^−2^	2.52 × 10^−2^	3.42	3.00 × 10^−2^	3.90	
	RM598	5	62.7	16,676,126	16,676,152	2.71 × 10^−3^	4.31	8.82 × 10^−3^	2.53 × 10^−2^	3.08	3.33 × 10^−2^	3.69	
	RM225	6	26.2	3,416,638	3,416,665	3.19 × 10^−3^	6.39	1.18 × 10^−2^	3.18 × 10^−2^	4.83	4.33 × 10^−2^	5.61	[38]
	RM234	7	93.9	25,420,132	25,420,157	2.47 × 10^−2^	5.61	3.53 × 10^−2^	4.88 × 10^−2^	5.09	5.00 × 10^−2^	5.35	
	*RM264*	8	138.2	27,926,633	2,7926,652	3.98 × 10^−3^	6.25	1.47 × 10^−2^	4.34 × 10^−3^	6.19	1.00 × 10^−2^	6.22	
	**RM512**	12	39.4	5,104,270	5,104,483	1.58 × 10^−2^	2.99	2.65 × 10^−2^	2.45 × 10^−2^	2.75	2.67 × 10^−2^	2.87	[39]

SSR markers in boldface represent the co-associated markers for PH, 2IN, 3IN and 4IN; SSR markers with underline represent the co-associated markers for PH, 1IN, 2IN, 3IN and 4IN; SSR markers in italics represent the co-associated markers for PH, 1IN, 3IN and 4IN; SSR markers in double-underline represent the co-associated markers for PL, 2IN, 3IN and 4IN. ^a^ The estimated map position (cM) and ^b^ physical position (bp) were inferred from the Gremene (“http://www.gramene.org/markers, accessed on 5 May 2021”) and NCBI (“http://blast.ncbinlm.nih.gov/Blast.cgi, accessed on 5 May 2021”) websites.

**Table 4 plants-11-00999-t004:** QTLs for PH and its component traits detected in XC-CSSL population.

Trait (cm)	Quantitative Trait Locus	Marker Locus Linked to the QTL	Marker Location (cM)	2018			2019		
				LOD	PVE (%)	Add (cm)	LOD	PVE (%)	Add (cm)
PH	*qPH-12*	RM19	20.9	2.96	17.77	15.10	2.52	17.19	14.85
PL	*qPL-7*	RM180	30.1	2.94	22.95	2.28	-	-	-
1IN	*q1IN-7*	RM180	30.1	4.95	28.42	5.31	-	-	-
	*q1IN-9*	RM257	79.7	3.20	16.92	5.68	-	-	-
2IN	*q2IN-7.1*	RM180	30.1	2.91	10.68	2.08	-	-	-
	*q2IN-7.2*	RM542	34.7	-	-	-	2.60	10.07	2.80
	*q2IN-12*	RM19	20.9	2.87	10.78	2.87	-	-	-
3IN	*q3IN-12*	RM19	20.9	-	-	-	2.57	16.70	2.82
4IN	*q4IN-8*	RM72	60.9	2.72	14.15	1.40	*-*	*-*	*-*

cM, centimorgan; PVE, percentage of phenotypic variance explained; Add, additive effect.

## Data Availability

The relevant data sources related to this manuscript are available within this paper and in the Appendix A.

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
