# Peer review of "Detection of QTLs for Plant Height Architecture Traits in Rice (Oryza sativa L.) by Association Mapping and the RSTEP-LRT Method"

_plants, 2022, doi:10.3390/plants11070999_

Round 1
Reviewer 1 Report
This manuscript deals with the dtection of QTL assocaited with plant archeticture of rice. The used methods are almost current.
Nevetheless this manuscript presents several concerns that should be addressed.
1-why authors did not present in introduction nor in discussion dwarfing genes in Rice? This is very important and may have a great influence on results discussion. This is truer since the means distribution seems bimodal which probably implies the presence of few genes involved in this distribution.
2-The presentation should be rewritten. some parts are redundants with tables (example L136-155). Moreover some sentences are are boring to read.
3- many studies have been done for Plant height genetic control in rice. Authors did not present clearly the originality of their study.
4-conclusion contains six lines. This study have been done on large populations during two years on many traits. It is possible to extend this conclusion with more potential interest of these results and perspectives.
Author Response
Responses to the comments of the Reviewer #1 on manuscript plants-1597885
Comments and Suggestions for Authors
This manuscript deals with the dtection of QTL assocaited with plant archeticture of rice. The used methods are almost current.
Nevetheless this manuscript presents several concerns that should be addressed.
Response: We appreciate your time, valuable suggestions and positive comment to improve scientific quality of our manuscript.
Point 1 - why authors did not present in introduction nor in discussion dwarfing genes in Rice? This is very important and may have a great influence on results discussion. This is truer since the means distribution seems bimodal which probably implies the presence of few genes involved in this distribution.
Response 1 – Thank you for your good comments. We have added the dwarf genes information in the introduction as well as in the discussion section of the revised manuscript.
Point 2 - The presentation should be rewritten. some parts are redundants with tables (example L136-155). Moreover some sentences are boring to read.
Response 2 - We have re-written the presentation and the redundant parts with tables have been expunged in the revised manuscript. Thank you.
Point 3 - many studies have been done for Plant height genetic control in rice. Authors did not present clearly the originality of their study.
Response 3 - Thank you for the comment. Although many studies have been done for plant height genetic control in rice, most of these studies were focus on the whole plant height. In this study, our focus is on the length of each internode of plant height, in order to shape an ideal plant height composition type that can increase plant height and biomass without lodging through genetic recombination, so as to achieve higher yield. In other words, it is the originality of this study to explore the favorable alleles of each internode that can design the ideal plant height. We have stated this idea in the introduction section.
Point 4 - conclusion contains six lines. This study have been done on large populations during two years on many traits. It is possible to extend this conclusion with more potential interest of these results and perspectives.
Response 4 - Thank you for your valuable observation. We have extended the conclusion in our revised manuscript with more potential interest of the results and perspectives.

Reviewer 2 Report
The authors of the manuscript “Detection of QTLs for plant height architecture traits in rice (Oryza sativa L.) by association mapping and the RSTEP-LRT method “ have performed an important research work using over 500 accessions of rice. There are some issues in the manuscript.
Minor issues
Check the format of references in the text
Such as “total phenotypic variation Zhu, et al. [22]”
Fig 1: statistical analysis is missing
Table 1,2, : statistical analysis is missing
Line 257 to 267: Unit of length (cm) is missing
Major comments
The manuscript is too lengthy. Authors can exclude unnecessary discussions in the RESULT SECTION.
Keep only important supplementary tables.
Important and recent work on rice in a similar field of study is still missing. Please see the recent works.
Author Response
Responses to the comments of the Reviewer #2 on manuscript plants-1597885
Minor issues
Point 1 - Check the format of references in the text
Such as “total phenotypic variation Zhu, et al. [22]”
Response 1 - I have deleted the word “Zhu, et al.” and all other references has been formatted accordingly in the revised manuscript.
Point 2 - Fig 1: statistical analysis is missing
Response 2 - Fig 1 showed a frequency distribution of plant height and its components. The descriptive statistics are listed in Table 1.
Point 3 - Table 1,2, : statistical analysis is missing.
Response 3 - Table 1 and Table 2 showed the descriptive statistics of the phenotypic values of the 6 traits studied. There was no statistically significant difference between 2018 and 2019 in means of the entire population in each trait.
Point 4 - Line 257 to 267: Unit of length (cm) is missing
Response 4 – Thank you. We have added “cm” in the appropriate sites in the revised manuscript.
Major comments
Point 1 - The manuscript is too lengthy. Authors can exclude unnecessary discussions in the RESULT SECTION.
Response 1 - We have deleted the unnecessary discussions in the RESULT SECTION. Thank you.
Point 2-Keep only important supplementary tables.
Response 2 - Thank you. Because the supplementary tables do not occupy the length of the article, we think the more detailed the better.
Point 3 - Important and recent work on rice in a similar field of study is still missing. Please see the recent works.
Response 3 - Thank you. Some important and recent work on rice in a similar field of study has been added into the INTRODUCTION SECTION and DISCUSSION SECTION of our revised version.

Round 2
Reviewer 1 Report
As authors have addressed all concerns in the manuscript, I'm satisfied by all modifications and propose to accept this manuscript in its present version.
Author Response
This blank page is because in this second round we don't have any issue from Reviewer 1.

Reviewer 2 Report
The author of manuscript QTLs for plant height architecture traits in rice (Oryza sativa L.) by association mapping and the RSTEP-LRT method" is important for plane science. THe are a few issues that required immediate correction and revision for safe journey.

Author Response
Responses to the comments of the Reviewer #2 on manuscript plants-1597885
Minor issues
The revision work of research article “Detection of QTLs for plant height architecture traits in rice (Oryza sativa L.) by association mapping and the RSTEP-LRT method” is satisfactory. The author has responded all the quieries and carefully added all the necessary information in the text. There are some issues in the text as given below.
Response: Thank you for your time, precious suggestions and positive comment to improve scientific quality of our manuscript.
Point 1 - Line 58-59: Adjust the following newly added information in the discussion section.
“The breeding of the majority of 58 semi dwarf rice varieties has been relying on genetic adoption of the mutant 59 alleles of SD1, which encodes a GA 20 -Oxidase, OsGA20ox2, for catalyzing 60 GA biosynthesis [7,8]”
Response – Thank you. We have adjusted the newly added information into discussion section (Please see Line 586-589).
Point 2 - Line 93-94: Repetition
“To the best of our knowledge, reports published on the elongated internode length traits in rice are limited, especially those of basal internodes….”.
Response 2 – We have expunged the repetition in Line 93-94 in the revised manuscript.
Point 3 - Line 124-125: “to date no report has documented a QTL analysis of the relationships between plant height and its components traits (panicle length and the length of all internodes) using two mapping strategies”.
Response 3 – We have kept the aforementioned information in the revised manuscript since its repetition in Line 93-94 has been deleted.
Point 4 - Figure 1 is not clear. Replace the figure with more distinct one.
Response 4 – We have improved and replaced the figure with more distinct one. Thank you.

Round 3
Reviewer 2 Report
The revision work of the manuscript "Detection of QTLs for plant height architecture traits in rice (Oryza sativa L.) by association mapping and the RSTEP-LRT method" is satisfactory. The author has responded to the queries and revised the manuscript carefully. Now, the manuscript may be accepted in the present form.